# DeepHandMesh-lite: Learning personalized hand shape using limited data and weak supervision

## Abstract

Being able to control the deformation of personalized, high-fidelity hand meshes in real-time contributes strongly to the feeling of presence in virtual reality. We present a method to learn an individual's hand shape based on 3D scans of the hand in different poses. For this, we rely on the data and hand shape model from the work of Moon et al. titled "DeepHandMesh" (DHM). We propose a novel algorithm to approximate hand joint pose based on joint position, and a loss function which leverages shape information contained in the silhouette. Of the 1070 high-resolution hand scans that DHM trains on in total, we choose only 24 poses representing primarily grasping scenarios. While the scans in DHM have been obtained with highly specialized equipment, our approach makes personalization of the hand mesh more feasible using limited resources. Our model is able to create subject-specific, posed meshes in real-time using joint positions as input, though there are sometimes artefacts visible in extreme poses that detract from the realism.

## 1 Introduction

As machines and humans work ever more closely in applications from healthcare to industry, it becomes more and more important to remove barriers in their interactions. Figuring out how to create seamless, realistic representations of a user's hands will play an important role in this. For example, the field of Human Activity Recognition requires large, realistic, often labelled datasets, which are onerous to acquire via recordings, but can be efficiently synthesized via 3D simulation (Zakour et al., 2021). Furthermore, in Virtual Reality (VR), a realistic, personalized hand appearance can increase a user's sense of embodiment (ownership of their virtual body) (Heinrich et al., 2021), which is important in industrial and clinical applications e.g. training, rehabilitation (Forster et al., 2022).

Creating hands with a realistic appearance is a challenging task, as anyone who has ever tried to draw hands can confirm. Reasons for this include the hand's many degrees of freedom of movement (Erol et al., 2007) and the complexities of self-collisions and deformations in the palm and fingers (Lee et al., 2006). Some of the current solutions for digital hand representations achieve a high degree of realism (e.g. LISA (Corona et al., 2022), NIMBLE (Li et al., 2022)) in the sense that an observer would judge their representations to look like real hands. Such approaches have their advantages, however they tend to rely on parameterised models of the hand, meaning that a specific subject's hand appearance can only be approximated based on what the model has learned from other examples of hands. They also typically require a lot of training data in order to generalize well. Other methods more focused on replicating an individual's hand appearance (e.g. DeepHandMesh (Moon et al., 2020), HandAvatar (Chen et al., 2023), LiveHand (Mundra et al., 2023)) still rely on large data sets or multi-view camera setups that are uncommon and expensive.

We instead propose a method to learn a real-time renderable, poseable and personalized representation of an individual's hand, with as realistic an appearance as possible, from a pared-down dataset consisting of 3D scans of a select number of hand poses. In particular, we rely on only 24 posed hand meshes from the DeepHandMesh dataset, which contains a total of 1070 such meshes. Note that joint position estimation is considered out of scope, i.e. accurate, labelled joint positions are

assumed to be available. Only one hand is considered. Optical properties such as diffuse colour, sub-surface scattering and specular reflection are neglected. Our contributions include an algorithm to approximate hand joint pose based on joint position, and a loss function which leverages shape information contained in the silhouette.

## 2 RELATED WORKS

Much recent research has focused on creating digital representations of real human hands, with a focus on problem areas including detecting hand poses, self-interaction and interaction with other objects, and 3D reconstruction of hands using images or joint positions as input. Some works hone in on a single one of these areas, while others create models that tackle multiple problems simultaneously. For our purposes, only works which create a 3D visualization of the hand's shape were considered.

One of the most influential recent hand models is MANO by Romero et al. (2017). Trained on thousands of hand scans from different subjects, their model takes hand shape and pose as input parameters and outputs a 3D hand mesh with pose-dependent deformations. Their use of Linear Blend Skinning and blend shapes makes the implementation efficient, but the low resolution of their hand mesh is a significant drawback. Several more recent mesh-based approaches are directly built on MANO (Qian et al., 2020; Zhang et al., 2021; Park et al., 2022; Karunratanakul et al., 2023). While some of these are still limited by a lack of realism due to low resolution (Qian et al., 2020; Zhang et al., 2021; Park et al., 2022), the problem is not insurmountable (Karunratanakul et al., 2023). Other mesh-based approaches which do not suffer from low resolution include DeepHandMesh by Moon et al. (2020) and NIMBLE by Li et al. (2022), though the latter still uses MANO for initialization.

A more realistic appearance, often including colour and texture, has been achieved by works that use a neural implicit representation of the hand (Karunratanakul et al., 2021; Corona et al., 2022; Chen et al., 2023; Lee et al., 2023; Mundra et al., 2023). This means that the hand's appearance is calculated as a continuous, implicit function approximated by a neural network. Some of these methods either build on MANO (Corona et al., 2022; Chen et al., 2023) or make use of MANO to improve their performance, for example by using MANO meshes for training (Karunratanakul et al., 2021) or as an initial estimate of hand shape (Mundra et al., 2023). The downsides of such implicit representations are that they require large amounts of data (multiple views) to learn a personalized appearance and can't easily be ported to common graphics applications (Karunratanakul et al., 2023). Additionally, the lighting and shadows from their training are baked into the model and difficult to change after the fact, e.g. by adding new lights (Chen et al., 2023; Mundra et al., 2023). For better immersion in VR, it is desirable for the solution to be renderable in various lighting conditions. As things stand, this encourages a mesh-based representation, rather than an implicit one.

Table 1 shows a comparison of some of the most recent methods on the basis of the features most important to our approach (namely personalized shape, mesh-based realisation, real-time performance and realism). All the listed methods use more training data than the limited set of poses we use. LiveHand (Mundra et al., 2023) performs well regarding all the aforementioned criteria, except that it uses an implicit approach. Only two methods meet all of our criteria: DeepHandMesh (DHM) (Moon et al., 2020) and HARP (Karunratanakul et al., 2023). HARP has the advantage of using monocular video as training data, which is easier to acquire than 3D scan data. However, while their hand looks good, any pose-specific deformations are a result of their approach being based on MANO. DHM, on the other hand, learns pose-based corrective shapes that are subject-specific, which is why we chose to adapt their method for our approach. We do however draw inspiration from HARP's use of the information contained in the silhouette, in order to improve our training loss. See Section 4.1 for more detail.

Table 1: Comparison of different 3D hand visualization methods. Note: '-' is used in the table when the relevant feature is not present, or its presence is not made clear in the corresponding paper

| Paper | Personalized | Mesh-based | Real-time | Realistic |
|---|---|---|---|---|
| DeepHandMesh (Moon et al., 2020) | ✓ | ✓ | ✓ | ✓ |
| HALO (Karunratanakul et al., 2021) | ✓ | ✗ | - | ✗ |
| LISA (Corona et al., 2022) | ✗ | ✗ | ✗ | ✓ |
| HARP (Karunratanakul et al., 2023) | ✓ | ✓ | ✓ | ✓ |
| HandAvatar (Chen et al., 2023) | ✓ | ✗ | - | ✓ |
| LiveHand (Mundra et al., 2023) | ✓ | ✗ | ✓ | ✓ |

## 3 METHOD

### 3.1 MODEL OVERVIEW

The proposed model builds on the decoder portion of DHM by Moon et al. (2020), as that paper is also concerned with realistic, custom 3D hand mesh generation. Their skeletal hierarchy and canonical hand mesh are used without modification, but their loss functions and training regime are adapted to the use of a drastically reduced pool of training data - we only make use of 24 out of the 1070 poses available.

The DHM decoder is trained to estimate the 3D hand mesh using the hand joint angles as input. While DHM trains its encoder to output these angles, we instead calculate them based on the 3D position coordinates of the main hand joints (referred to as keypoints) - see Appendix A for details on which joints are included, and Section 3.3 for how the angles are calculated. The information on which hand and which joint each keypoint corresponds to is also provided to the model.

As shown in Figure 1, for a given input pose, we first estimate the normalized pose matrices $\mathbf{P}_j$ of each joint $j$ in the skeleton using the algorithm described in Section 3.3. The term 'normalized' here refers to the fact that the joint locations are adjusted so that regardless of input, the lengths of the hand bones remains constant (equal to the length of the training subject's bones) across all poses.

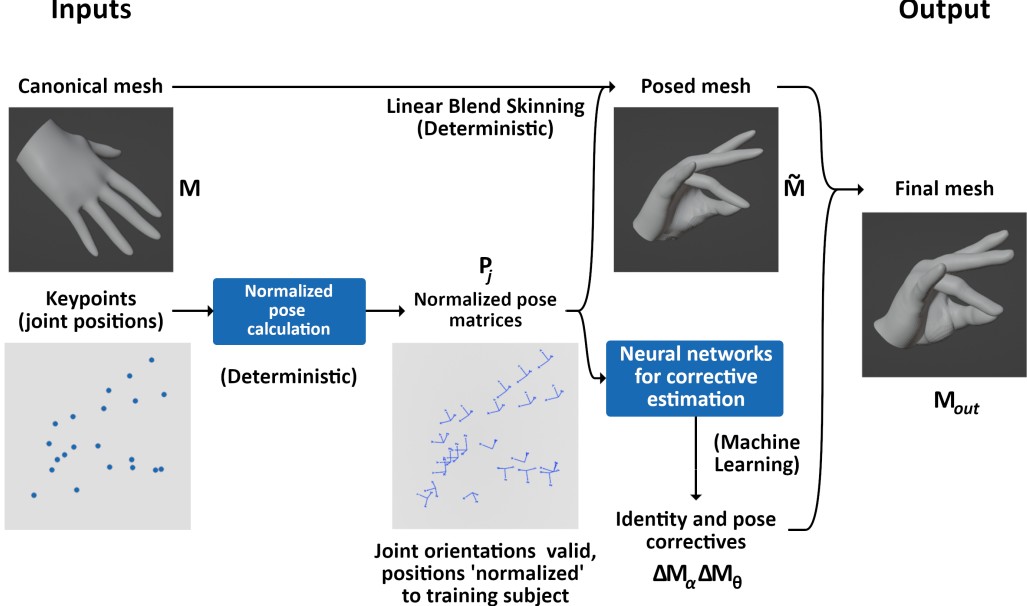

Figure 1: An overview of the proposed model

This means that the trained hand model can be driven by joint position data from any other person. These normalized pose matrices are then used to pose the canonical mesh $\mathbf{M} \in \mathbb{R}^{V \times 3}$ using Linear Blend Skinning (LBS), as described by Lewis et al. (2000).

Next, the trained model estimates the following additive local vertex position correctives:

- Identity-based corrections $\mathbf{\Delta M}_\alpha \in \mathbb{R}^{V \times 3}$: uses a neural network with 2 fully connected layers and a hidden activation size of 256. The input is a normally distributed random vector $\alpha \in \mathbb{R}^{32}$ which uniquely identifies the current subject.

- Pose-based corrections $\mathbf{\Delta M}_\theta \in \mathbb{R}^{V \times 3}$: uses a neural network with 2 fully connected layers and a hidden activation size of 256. The input is the current pose $\theta \in \mathbb{R}^{4N_j}$, where $N_j$ is the number of joints in the skeleton, i.e. $\theta$ is a stacked vector of joint orientation vectors, where each orientation is represented using canonized quaternions.

The output mesh is given by:

$$\mathbf{M}_{out} = \tilde{\mathbf{M}} + \mathbf{\Delta M}_\alpha + \mathbf{\Delta M}_\theta \tag{1}$$

where $\tilde{\mathbf{M}}$ is the canonical mesh after the calculated pose has been applied.

## 3.2 DATA CURATION

The following data are required in order to train the model: a) detailed, renderable 3D-scanned hand meshes in a variety of poses - rendered into depth maps which are used as ground truth data for learning the hand shape, b) corresponding 3D keypoints (joint locations), information on handedness (left/right), and which joint each keypoint corresponds to. These are used to calculate the hand pose.

In the early stages of this work, an approach to data acquisition involving a hand-held 3D scanner was explored. While it was possible to capture very fine detail from a live subject, it proved very difficult to acquire a complete scan of the hand without significant artefacts due to motion of the subject. The proposed solution was to create a cast of the hand which could then be scanned without having to worry about the subject's motion.

The creation of such a dataset was out of scope for this paper, so instead, the decision was taken to choose distinct poses from an existing dataset - in this case, the DHM dataset - as if they had been created under the described conditions. This entailed selecting a limited number of hand poses according to the following parameters: a) The number of poses should ideally be low, as creating a cast of each pose is time consuming and impractical for hundreds of poses (as required by a model like DHM), b) The poses should be representative of the required range of motion for all hand joints. We focus on poses related to object grasping, rather than abstract, unusual hand poses.

According to these criteria, of the 1070 training scans of a single subject provided by DHM, only 24 poses were used (see Appendix A). The DHM dataset also includes the corresponding keypoints, which were obtained via multi-view 3D hand pose estimation, referencing the work of Li et al. (2019).

## 3.3 CALCULATING THE HAND POSE FROM 3D KEYPOINTS

To pose the canonical hand mesh using LBS, both the positions and orientations of the hand joints need to be known. DHM learns the orientations directly from images. But since we use joint locations as inputs and only a limited number of poses, learning the joint orientations was not feasible for this work. Instead, the orientations are deterministically calculated based on the knowledge of human hand anatomy and the provided joint positions as described in the next two paragraphs.

The planes of rotation of the various joints can be consistently approximated regardless of identity or pose. This can be done via a calculation based on the direction vectors of the associated bones (first plane vectors) and the positions of the thumb carpometacarpal (CMC) and all metacarpophalangeal (MCP) joints (second plane vectors). This is shown in Figure 2. The second plane vectors are calculated as follows, using the vectors $v_i$ with $i \in \{1, ..., 12\}$ which correspond to the arrows labeled in Figure 2: $v_{Thumb} = v_9 \times v_{11}$; $v_{Index} = v_4 \times \frac{v_7 + v_8 + v_{12}}{3}$; $v_{Middle} = v_3 \times \frac{v_6 + v_7}{2}$; $v_{Ring} = v_2 \times \frac{v_5 + v_6}{2}$; $v_{Pinky} = v_1 \times v_5$. These were formulated to try and emulate the rotation

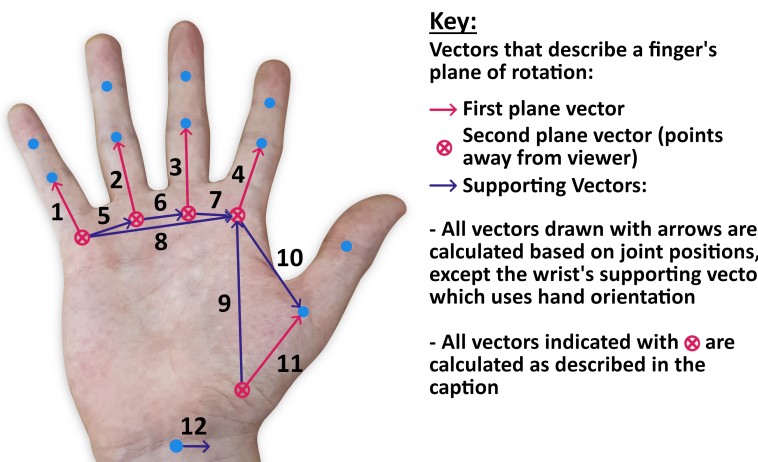

Figure 2: Vectors used to describe finger rotation planes.

planes dictated by hand anatomy, and were chosen such that they gave good approximations across all training poses.

When a joint is bent far enough, the plane of rotation can be directly calculated instead of approximated. This is done by assembling the (column) position vectors of the current joint, its parent and its child into a $3 \times 3$ matrix. Then, using Singular Value Decomposition (SVD), the normal vector of the plane passing through the three points can be found by taking the third column of the left singular vectors. When this calculation is not possible (i.e. when joints of a finger are collinear), the parent joint's rotation plane - approximated as described under the previous point - is used.

By combining direct calculation and anatomy-based estimation on a joint-by-joint basis, the orientations of all hand joints can be calculated in a consistent way. Specifically, in keeping with the canonical hand model, the joints of each digit are first oriented so that their negative x-axes point directly at their respective child joints. Then they are rotated on their newly positioned x-axes, so that their y-axes lie in the respective calculated rotation planes.

Note that there is a certain degree of imprecision in the the (estimated) input keypoint positions. This results in the lengths of some bones being slightly different in different poses, which doesn't occur in nature. DHM addresses this by learning the bone lengths. For the same reasons outlined above with regard to poses, we instead calculate the average length of each bone across the 24 poses, and then adjust the joint positions so that the average bone length is used across all estimated poses. This results in a kind of normalization such that after training, keypoints from any subject's hand will be translated into hand poses suited to the subject on whom the model was trained.

The final orientations are defined as rotation matrices for the purposes of posing the canonical hand mesh, and converted into a quaternion representation for predicting the pose-based vertex position corrective.

## 4 TRAINING

### 4.1 LOSS FUNCTIONS

The total loss function to be minimized is composed as follows:

$$L = L_P + L_{DM} + \lambda L_L \qquad (2)$$

Where:

- $L_P$ is the penetration loss, which accounts for interactions of the hand with itself, penalizing mesh penetration
- $L_{DM}$ is the depth map loss, which penalizes differences between the shapes of the estimated and ground truth hand meshes

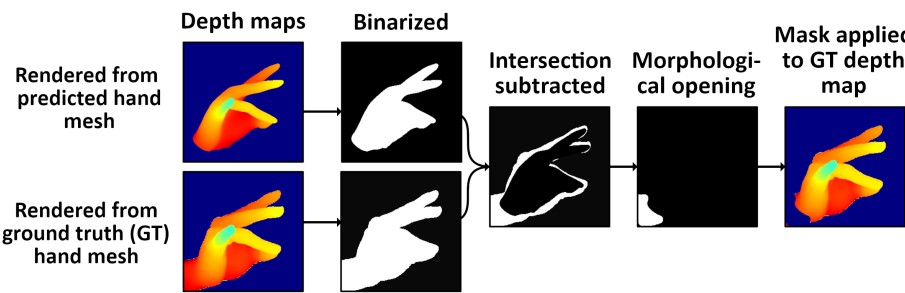

Figure 3: Masking out the wrist using image processing

- $L_L$ is the Laplacian loss, which penalizes deviations from the local structure of the hand mesh based on its topology, acting as a regularizer

- $\lambda = 5$ is the shrinkage coefficient for the regularization

$L_P$ and $L_L$ are used as defined in Moon et al. (2020). $L_{DM}$, on the other hand, is significantly altered.

The inputs of $L_{DM}$ for a given pose are $256 \times 256$ pixel depth maps of the ground truth hand scan and the estimated hand mesh, both rendered from the same camera view (using Neural Renderer (Kato et al., 2017), so that they are loss differentiable). Because the ground truth hand scans include the forearm and wrist, while the canonical/estimated meshes do not, the difference between the two maps will always contain a large error which is affected more by relative camera angle than hand pose. DHM solves this by only comparing those pixels where both depth maps contain foreground information (i.e. where both ground truth and estimated pixels belong to the hand). However, this fails to make use of information contained in the deviation between the ground truth and estimated silhouettes for a given pose.

One way to solve this would be to add a forearm to the canonical mesh and train its appearance as well, as in Karunratanakul et al. (2023). However, this would require extending the skeleton and having the keypoint estimation estimate additional joint positions. Instead, we chose to mask out the unwanted forearm and wrist pixels from the ground truth depth maps, visualized in Figure 3.

In short, both maps are converted into binary maps where all hand pixels are white and all background pixels are black. Then the intersection of both maps is subtracted from the ground truth binary map and a morphological opening operation is performed, with a $13 \times 13$ pixel kernel. This removes all remaining parts of the ground truth map except the unwanted wrist/forearm area. This area is now isolated and can be masked out of the ground truth depth map, allowing a direct comparison between depth maps while taking the silhouettes into account. Doing so noticeably improved the speed and accuracy of training.

## 4.2 IMPLEMENTATION DETAILS

We train our model using DHM's canonical hand mesh, which has $12553$ vertices. The neural networks for estimating shape correctives are realised using PyTorch by Paszke et al. (2019) and the Adam optimizer by Kingma & Ba (2017). The batch size is set to 8, meaning that 8 different training poses are trained in parallel. During training, each pose is rendered from 60 different random viewpoints, and the resulting depth maps are compared to the corresponding ground truth depth maps.

The dataset is duplicated such that at the end of an epoch, each of the 24 poses has been trained 3 times (i.e. there are 72 training poses in total, so 1 epoch = 9 batches). As the order of the training examples is reshuffled every epoch (and because of random viewpoint selection), the results of the training are not deterministic. After a trial of 200 epochs with a learning rate of $0.001$ using an NVIDIA GeForce RTX 2070 GPU, the overall loss was shown to be minimal at around 25 epochs.

Figure 4: Depth map loss for the test sequence. The average across views is plotted. The four best (green) and worst (red) frames are marked with vertical lines. The frame numbers correspond to video frame numbers from the DeepHandMesh dataset capture process Moon et al. (2020).

## 5 RESULTS

To evaluate the performance of the model and its ability to generalize, it was necessary to test it using previously unseen/untrained poses. The ideal case would have been to test using data similar to the intended application - a continuous sequence of grasping/interaction poses. However, such a sequence was not available for testing, so instead, 11 previously ignored poses from the DHM dataset were used. For each of these 11 poses, ground truth 3D meshes are constructed from a short video sequence that goes from a rest pose to the pose in question and then back. Only every 6th frame from each such sequence is available, resulting in 148 ground truth meshes used for testing.

To gauge the model's performance, only the depth map loss $L_{DM}$ was considered, as it directly describes how closely the model's output resembles the ground truth data. The view losses across all testing poses are calculated for four different camera views (top, bottom, left, right). These four view losses are averaged and plotted in Figure 4. Also plotted are vertical lines for the four best results (in green) and the four worst results (in red), as well as the overall average testing loss after 24 epochs (see figure title).

During testing, the decoding step which outputs the refined hand mesh took on average 13ms, which allows for a theoretical frame rate of around 76fps. This leaves about $20ms$ per pose in which to acquire input keypoints and render the output mesh, if a frame rate of 30fps is to be achieved.

In order to visualize the results, the model output ($D_m$) and ground truth ($D_{gt}$) depth maps as well as the error ($D_e = D_{gt} - D_m$) between them are colourized and displayed next to each other for each pose in the testing data. For an example, see Figure 5, where the four best and four worst results are visualized.

## 6 DISCUSSION

A visual inspection of our results (see Figure 6) shows that the output mesh closely resembles the ground truth, although some details are incorrect or missing. For example, the finger and thumb lengths aren't quite correct, and structures like the tendons on the knuckles are merely suggested, while finer details, like folds in the thumb area, are missing. When regarding the test visualizations (Figure 5), the better results come close to matching the silhouettes of the reference scans. The general shape of the digits in particular seems to match the model quite well. However, even among the best results, and particularly among the worst, the biggest deviations occur around the knuckles and at the base of the thumb.

For a more quantitative comparison, we trained the DHM model on all available poses. We then tested the model using the same methodology that was used to test our approach. When using our calculated joint poses and only the decoder part of the DHM model, the overall average loss was worse than ours (21.381 vs 17.856). The reason for this appears to be that their encoder and decoder are trained in tandem, and our method for calculating the hand joint poses produces different results to their encoder. When using both the encoder and decoder of the DHM model (i.e. starting with an image as input), the overall average loss was again worse than ours (20.468 vs 17.856). In this case, it was because their encoder performs worse at joint pose estimation than our approximative method.

Note that in both of the above cases, despite scoring worse than our results due to lower accuracy, the DHM results appear somewhat more natural. The artefacts seen around the knuckles in our results

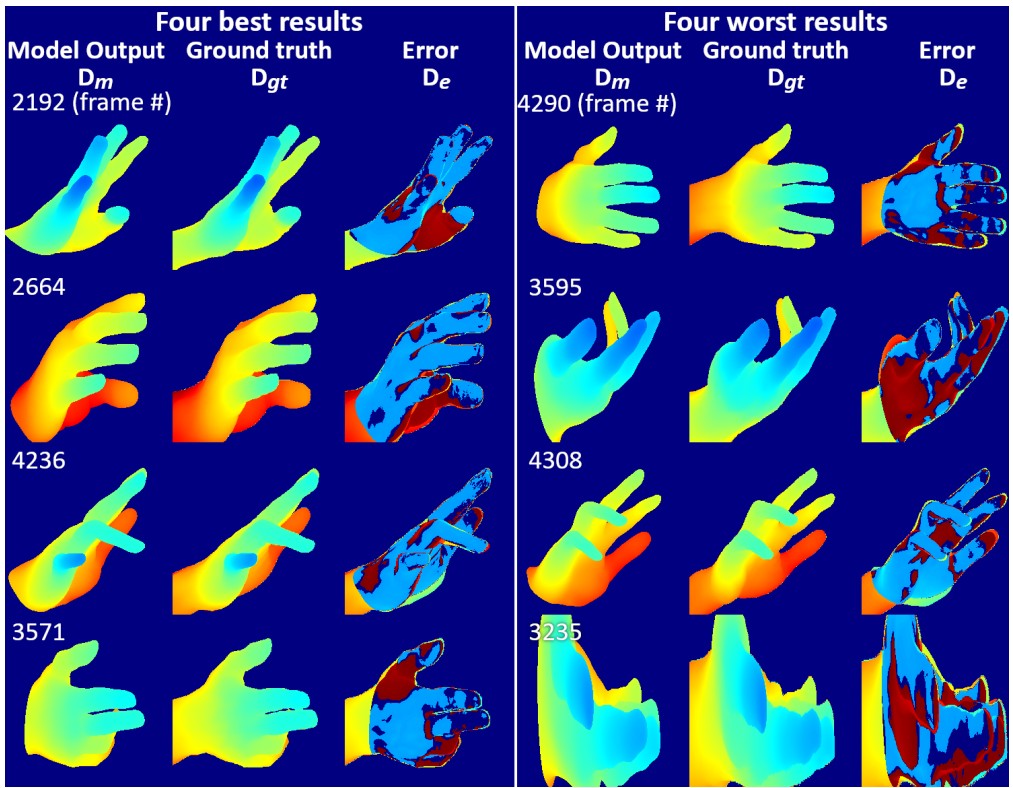

Figure 5: Visualization of the four best (left) and four worst (right) results. Both sides show the results in order of descending quality i.e. the overall best result is the top left, the overall worst the bottom right. The frame number is shown in the top left of each visualization.

are absent. Note also that DHM had already seen the testing poses during its training. To account for this, we also tested our approach and theirs on our 24 training poses, but it didn't significantly alter how they performed.

Since pose estimation is not in the scope of our approach, we also trained only the DHM decoder using our joint poses as input. After 18 epochs of training, their model performed slightly better than ours. However, artefacts, were present especially around the base of the thumb, where there were none in the standard DHM results. This indicates that the artefacts potentially arise due to our method of joint pose approximation.

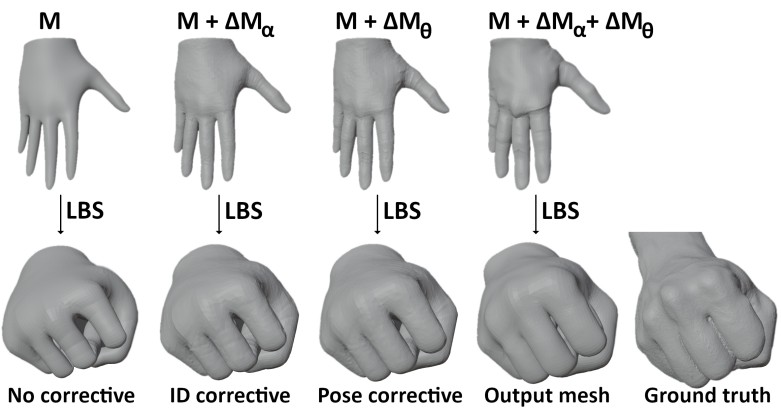

Figure 6: A summary of the different model outputs

The precise reasons for this are unclear. Large errors or inconsistencies in the calculated joint poses would lead to problems with pose-based corrective estimation, but to our knowledge such errors are small, at least for our training poses. These problematic regions (knuckles, base of thumb) are areas where a lot of deformation occurs, and it is possible that our pose approximation method, which uses only anatomical joints, lacks sufficient degrees of freedom to be able to properly learn and untangle the pose and identity based correctives. DHM's model includes an additional (non-anatomical) thumb joint, which may explain the difference in results. Lastly, it is worth mentioning that some of the ground truth scans contain artefacts such as noisy vertices and incomplete regions due to occlusions or reconstruction errors. These certainly affect model performance adversely, especially with so few training examples.

## 7 CONCLUSION

### 7.1 SUMMARY

We trained a model to create a realtime-poseable facsimile of a subject's hand from scans of 24 select poses. Our specific contributions include an algorithm to estimate hand joint pose based on joint position, and a depth map loss function that ignores unwanted parts of the ground truth data while taking useful areas of the silhouette into account. Furthermore, we optimised the learning rate and number of camera views used for training under the new conditions.

The resulting model is able to produce meshes that resemble the target, however, not without noticeable errors, which detract from the overall realism. These errors are most pronounced in regions of high deformation e.g. the knuckles and the base of the thumb. Otherwise, the level of detail captured is fairly low e.g. the mere suggestion of tendons/veins. Finer details such as folds - even those which the mesh has enough resolution to represent - are not captured well.

### 7.2 FUTURE WORK

Apart from obtaining and using cleaner ground truth scans, some other steps could be taken to improve the poor results in areas of high deformation. One reason these areas suffer may be because a single pose vector is used to describe the entire hand. The pose vector has 88 dimensions (22 joints, 4 quaternion values per joint) and there are only 24 training poses. This means that training poses may be very far apart in pose space, making a generalization or interpolation between poses less accurate. To address this, we can separate the correctives into regions and train individual models for each region. In this way, one might be able to compensate for the lack of training data and maximize its usage by incorporating knowledge of the model. For example, the interphalangeal joints of the pinky do not affect thumb deformations, so they can be explicitly excluded from training thumb shape correctives without loss of information. A limitation of this approach would be that subtle changes due to motion of joints outside the defined region would be lost.

Another possibility is that the skinning of the canonical mesh is sub-optimal for the subject's particular hand shape and deformations. A solution could be to learn skinning weight correctives alongside the vertex position correctives. Additionally, the limitations of LBS may play a role. Different approaches like Dual Quaternion Skinning (Kavan et al., 2007), spline-based skinning (Forstmann & Ohya, 2006) or Implicit Skinning (Vaillant et al., 2013) may be better suited.

Additionally, one could investigate more thoroughly which set of 24 poses are most useful for training a model like this. In this work, the decision was made based on visual inspection, but a RanSaC-like approach might be more robust. Testing the model with additional subjects would also be instructive.

## 8 REPRODUCIBILITY

Section 4.1 provides details of our novel depth based loss function, whereas Section 4.2 provides implementation details. Furthermore, We will provide a link to an anonymous code repository on the discussion forum that will be opened for our submitted paper, by making a comment directed to the reviewers and area chairs.

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

## A    APPENDIX

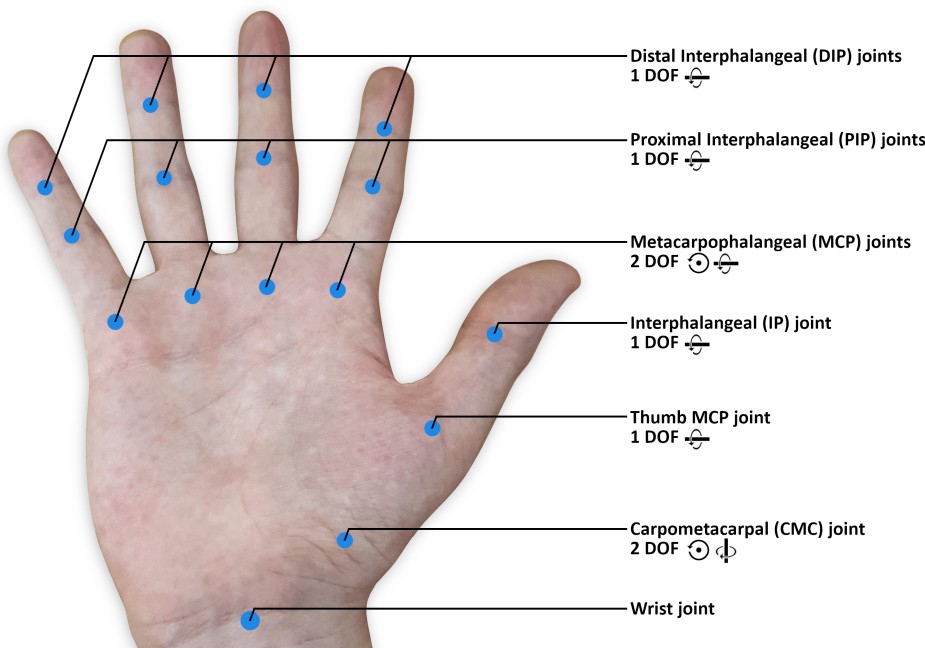

Figure 7: The main joints of the hand, which correspond with keypoints in the training data. (DOF = Degrees of Freedom)

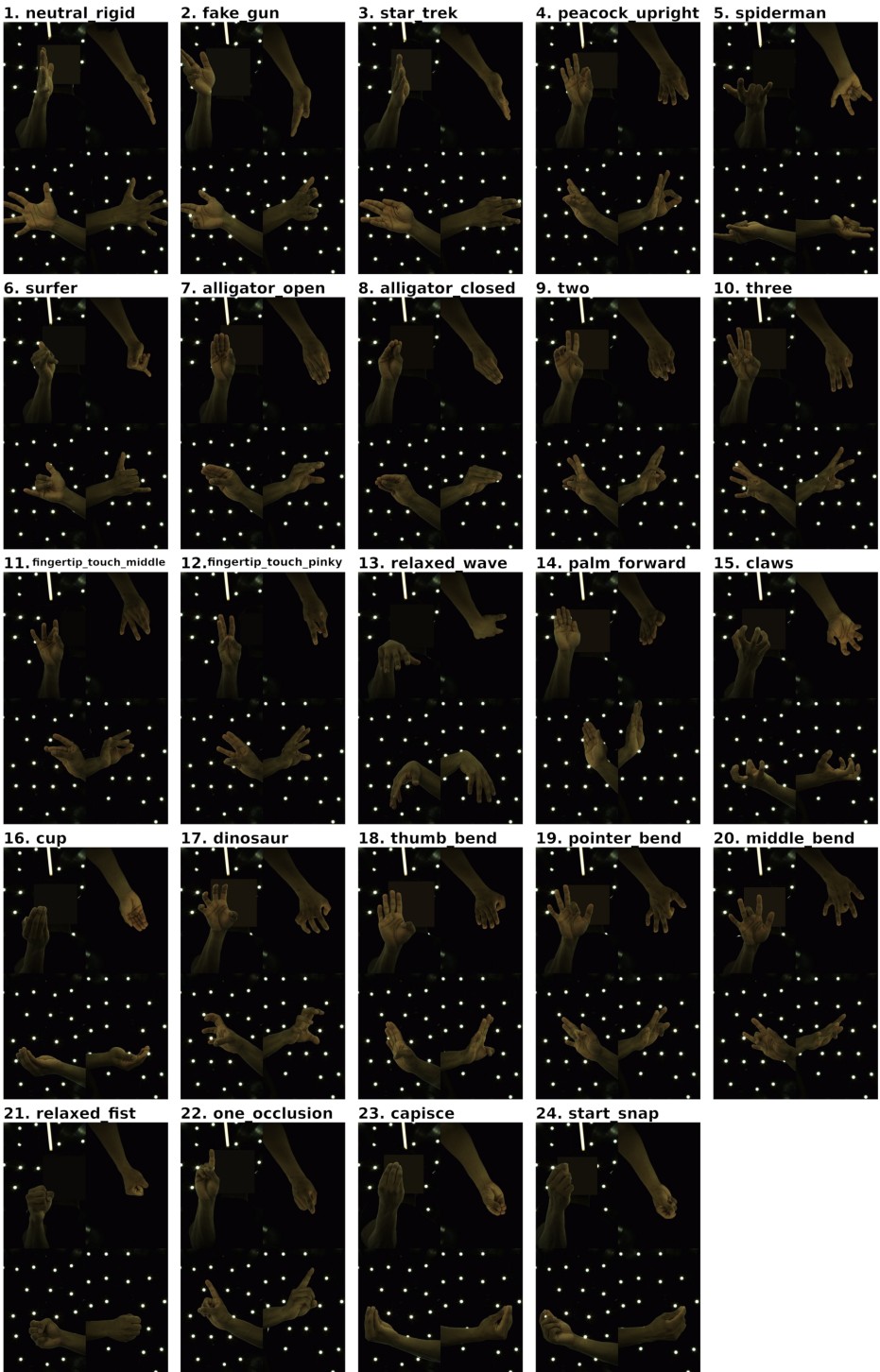

Figure 8: The 24 poses selected for training. The images are taken from the DeepHandMesh dataset Moon et al. (2020).

