# OpenReview forum: "DeepHandMesh-lite: Learning personalized hand shape using limited data and weak supervision"
_ICLR.cc/2024/Conference — ICLR 2024 Conference Withdrawn Submission_

### Official Review · Reviewer_unTW · 2023-10-29

**Soundness:** 2 fair
**Presentation:** 3 good
**Contribution:** 1 poor
**Rating:** 3
**Confidence:** 5

**Summary:**

This paper describes a method for creating a custom, identity-preserved 3D hand mesh with pose from given 3D joints. Instead of commonly used MANO model. This paper proposes to train a model to regress the detailed vertex offset to achieve more realistic and custom effects. To achieve this, the proposed model need to train a model to learn from 3D hand scans of 24 select poses (along with their 3D poses). In this way, the model is supposed to determine the pose-related & identity-related vertex offsets and deformations. The proposed method is evaluated on a small set of ground-truth meshes from DHM dataset.

**Strengths:**

1. The idea of training a model to learn the pose-related & identity-related vertex deformation is interesting.
2. The presentation is clear and easy to understand.

**Weaknesses:**

1. Evaluation is only performed on a very limited small set of data. The precision & generalization ability hasn't been not fully evaluated.
2. Lacking comparisons with traditional methods, like modeling pose-related deformation via statistical analysis.

**Questions:**

The idea is interesting. But due to data lacking problem, the evaluation part is not very convincing. Is there is a way to collect / generate more 3D hand scans for training & evalution?

---

### Official Review · Reviewer_Z5hG · 2023-10-30

**Soundness:** 1 poor
**Presentation:** 2 fair
**Contribution:** 1 poor
**Rating:** 3
**Confidence:** 4

**Summary:**

This paper presents a method for personalized hand mesh modeling from a small number of hand scans. The proposed method is built upon DeepHandMesh, an existing method that learns hand mesh reconstruction in an encoder-decoder fashion. The difference is that DeepHandMesh learns to reconstruct hand mesh from image inputs, while this paper outputs hand mesh given only joint position. To this end, this paper designs an inverse kinematic algorithm to compute joint rotations from joint positions. Results show that the proposed method is able to learn plausible hand model given only 24 hand scans.

**Strengths:**

The proposed method is able to learn plausible hand model given only 24 hand scans. It supports cross-identity animation since only 3D joint positions are the only input.

**Weaknesses:**

* The quantitative evaluation is weak. The authors only report the training loss for DHM and their method, but this is not a valid metric since it has been used for network training. I think more metrics should be consider. For example, at least 3D joint distance errors and mesh vertex errors should be reported as is done in DeepHandMesh.

* The comparison against existing method is not convincing enough. The authors only compare with DeepHandMesh, which is published in 2020 and is not the state-of-the-art at present. I think the authors should compare with the methods listed in Table 1 to make this paper stronger.

* The modification to depth map loss, $L_{DM}$ is trivial and I don't think it can be regarded as a technical contribution. In addition, I do not found any ablation study to support the effectiveness of this modification.

* In Introduction, the authors claim that the proposed method can learn the realistic appearance of hands, but I cannot find any demonstration or experiments to support this claim.

* In Sec 3.2, the authors mention a challenge about data collection. But in the end, the authors do not provide any solution for this challenge. Instead, "the decision was taken to choose distinct poses from an existing dataset". I feel confusing reading this paragraph.

**Questions:**

See [Weaknesses].

---

### Official Review · Reviewer_YMSw · 2023-10-31

**Soundness:** 2 fair
**Presentation:** 2 fair
**Contribution:** 2 fair
**Rating:** 3
**Confidence:** 2

**Summary:**

This paper works on hand mesh shape reconstruction from 3D scans of different poses. Built upon DeepMeshHand (DMH), this paper can reduce the required 3D scans number from 1070 to 24, and still achieve comparable performance even with such limited resources. Compared to DMH, they propose a direct calculation to approximate plane rotation instead of using network fitting. Besides, they utilize morphological opening operation to remove wrist areas while keeping silhouettes into loss optimization, which solves the ineffectiveness existed in the depth map loss function. They compared with DMH on shape reconstruction on loss metric and analyze the left limitations in this work.

**Strengths:**

- They consider a more challenging scenario to recover high-fidelity hand meshes, where only limited 3D poses are available (i.e., 24 poses compared with previous 1070 poses).
- Their method is easy to understand.

**Weaknesses:**

- This paper is not well organized and hard to follow. For example, the dataset information is excluded that needs to refer to other literature, the method section looks like simply introduces their pipeline without rational explanation, and Sec.2 is too long and needs to be divided into subtitles.
- I think the contributions proposed in this paper are hard to meet the accept criteria of ICLR.
- The method they proposed is like a combination and lacks coherence in rationality.

**Questions:**

- [Pose Calculation from 3D Keypoints] The reason why the authors propose a new way to calculate plane rotation is unexplained. Considering there is already one literature [add Spurr et al. 2020] to calculate joint rotation, I suggest the authors compare with it to highlight the difference and necessity.
- [Depth Map Loss] I think the revised depth map loss, as a technical improvement, is not yet important as an outstanding contribution. Simply using an auxiliary network trained on hands with color gloves can also have great segmentation to remove wrist/forearm area [add: Bojja et al. 2019]
- In Sec.6, this paper uses depth map loss to evaluate the performance, which is different from previous work using mesh vertex error. There is no specific explanation provided in the paper, what kind of depth map loss do they use? Besides, I would question the rationality of using loss as an evaluation metric here, especially when this paper is based on the modified depth loss term for training.
- Can the authors provide more explanation on why the selected poses are considered as grasping? And why grasping poses are better than others? I believe that providing further clarifications would enhance the comprehension of the fundamental principles underlying their method.
- The experiment setting discussed in Sec.5 is unclear. How does 148 derive? Besides, I would suggest the authors try to add experiments on other datasets, such as HO-3D, Dex-YCB (grasping scenarios) [add: Hampali et al. 2020, Yu-Wei et al. 2021], or manually annotate some real-world examples, to prove the effectiveness of the method. I think such comparisons will be more convincing.
- In DHM, it also compares with the sotas under similar mesh resolution and datasets based on MANO. I suggest the authors add these comparisons as well, as the proposed method should also work on low-resolution mesh, which can verify the method’s effectiveness.

(add: Bojja et al. 2019) Handseg: An automatically labeled dataset for hand segmentation from depth images, CRV’19.

(add: Hampali et al. 2020)Honnotate: A method for 3d annotation of hand and object poses, CVPR’2020

(add: Spurr et al. 2020) Weakly supervised 3d hand pose estimation via biomechanical constraints, ECCV’2020

(add: Yu-Wei et al. 2021) DexYCB: A benchmark for capturing hand grasping of objects. In CVPR’2021